# A Facile Approach to Solid-State White Emissive Carbon Dots and Their Application in UV-Excitable and Single-Component-Based White LEDs

**DOI:** 10.3390/nano9050725

**Published:** 2019-05-10

**Authors:** Xiangyu Feng, Kai Jiang, Haibo Zeng, Hengwei Lin

**Affiliations:** 1MIIT Key Laboratory of Advanced Display Materials and Devices, Institute of Optoelectronics & Nanomaterials, School of Materials Science and Engineering, Nanjing University of Science and Technology, Nanjing 210094, China; fengxiangyu@nimte.ac.cn; 2Key Laboratory of Graphene Technologies and Applications of Zhejiang Province & Ningbo Institute of Materials Technology & Engineering (NIMTE), Chinese Academy of Sciences (CAS), Ningbo 315201, China; jiangkai@nimte.ac.cn

**Keywords:** carbon dots, photoluminescence, solid state emission, white light-emitting diodes

## Abstract

Large-scale applications of conventional rare-earth phosphors in white light-emitting diodes (W-LEDs) are restricted by the non-renewable raw material sources and high energy consumption during the production process. Recently, carbon dots (CDs) have been proposed as promising alternatives to rare-earth phosphors and present bright prospects in white lighting. However, the use of CDs in W-LEDs still has two major obstacles, i.e., solid-state quenching and lack of single-component white emissive products. In this work, a facile, rapid, and scalable method for the preparation of solid-state white emissive CDs (W-CDs) is reported via microwave-irradiation heating of L-aspartic acid (AA) in the presence of ammonia. The W-CDs exhibit blue photoluminescence (PL) in dilute aqueous dispersion and their emission spectra gradually broaden (emerging new emissions at orange-yellow regions) with concentration increases. Interestingly, the W-CDs powder displays a very broad PL spectrum covering nearly the whole visible-light region under ultraviolet (UV) excitation, which is responsible for the observed white emission. Further studies revealed that the self-quenching-resistance feature of the W-CDs is probably due to a covering of polymer-like structures on their surface, thus avoiding the close contact of nanoparticles with each other. PL emission of the W-CDs is reasonably ascribed to a cross-linked enhanced effect (CEE) of the sub-fluorophores contained in the material (e.g., –NH_2_ and C=O). Finally, applications of the W-CDs in fabricating single-component-based W-LEDs using commercially available UV chips were attempted and shown to exhibit satisfactory performances including high white light-emitting purity, high color rendering index (CRI), and tunable correlated color temperature (CCT), thus rendering great promise for W-CDs in the field of white lighting.

## 1. Introduction

The ever-growing interest and significant progress of carbon-based nanomaterials have witnessed broad prospects in the fields of sensing, catalysis, biomedicine, and optoelectronic devices [1,2,3]. As a prominent representative among numerous carbon-based nanomaterials, carbon dots (CDs) have attracted intense academic attention due to their superior features, such as low-cost and abundant raw materials, facile and manifold synthetic routes, high stability against heat and radiation, and benign biocompatibility [4,5,6,7,8]. In comparison to the conventional light-emitting materials (e.g., rare earth phosphors and semiconductor quantum dots (SQDs)) [9,10,11,12,13], solid-state emissive CDs hold many advantages. However, one of the most prominent problems that has hindered wide applications of CDs is their drastic photoluminescence (PL) quenching in the solid state, well-known as aggregation-caused quenching (ACQ) [14,15]. Consequently, the excellent PL properties of CDs in the dispersion state cannot be usually directly employed in solid-state conditions.

In order to conquer solid-state PL quenching of CDs, several solutions have been proposed. First, solid matrices, such as starch [16], barium sulfate [17], silicates [18,19], silica gel [20], mesoporous silica [21], cage-like molecules [22], and polymers [23] were adopted to disperse CDs, so that the distances of CDs could be separated by a reasonable value from each other, thus avoiding the resonance energy transfer (RET) process and/or direct π-π interactions. Second, long-chain-contained molecules were used as carbon sources, such as poly(vinyl alcohol) (PVA) [24,25], KH-792 [26,27], organo-functional silane [28] and Tween 80 [29], which endowed the produced CDs by covering the long chain structures on their surface and led to effective resistance of ACQ. Finally, preparation of CDs using sub-fluorophores (e.g., –NH_2_ and C=O) containing small molecules as carbon sources via polymerization and carbonization processes [30,31]. The sub-fluorophores in CDs were speculated to act as main emissive centers, and their intrinsic weak PL performance could be drastically enhanced due to the crosslink enhanced effect (CEE) [32,33]. Since the sub-fluorophores did not suffer from direct π-π interactions and were protected by the polymer frameworks, the effective resistance of solid-state quenching could be realized [34]. Although these above strategies satisfactorily solved solid-state PL quenching of CDs, the direct obtained solid-state white emissive CDs are still highly challenging and have very rarely been reported so far [26].

One of the most impressing applications of solid-state emissive CDs is to fabricate white light-emitting diodes (W-LEDs) [24,35]. Current commercial white lighting is generally obtained by a combination of blue-emitting InGaN chips and yellow-emitting rare-earth phosphors [36,37,38]. Although such phosphors are highly efficient light conversion materials, they suffer from expensive resources and high energy consumption during the production process. SQDs, another kind of commonly used phosphors, however, are harmful to both the human and the eco-system due to the presence of highly toxic heavy metals [11,12,13,39,40]. Currently, the majority of reported CDs-based W-LEDs can be divided into two categories, either encapsulating yellow emissive CDs on blue-emitting chips [41,42], or assembling multiple CDs with diverse emission, typically red, green and blue, on ultraviolet (UV) chips [43,44,45]. In the former case, since the emission intensity of blue chips varies with the working currents [45], stable correlated color temperature (CCT) and color purity are hardly maintained. Moreover, the lack of orange-red emission would result in a low color rendering index (CRI). The latter strategy could overcome these problems, but it is relatively complicated and difficult to encapsulate multiple CDs (appropriate ratios among the CDs have to be carefully screened), and meanwhile a constant white light emittance could hardly be maintained due to the different emission decays of the used CDs. Therefore, the use of single-component white emissive CDs that can resist ACQ and be excitable by UV light is much more preferable.

Up to date, only a handful of white emissive CDs have been reported, exhibiting either ACQ property (i.e., emitting white light only in dilute dispersions) or very low PL quantum yields (QYs) in the solid state [14,46,47,48,49]. In this work, a very facile and rapid method was developed to prepare white emissive CDs (named W-CDs). Importantly, the W-CDs powder was observed to be resistant to quenching and emitted white-light under UV excitation with a respectable PL QY of 6.7%. Furthermore, the possible formation process, chemical structure, and PL mechanism of the W-CDs in the dispersion state and solid state are discussed. Based on the superior solid state white emissive feature, the W-CDs were applied to fabricate single-component-based W-LEDs using commercially available UV chips (365 nm) and presented satisfactory performances.

## 2. Materials and Methods

### 2.1. Reagents

l-Aspartic acid (AA) and l-glutamic acid were purchased from Aladdin (Shanghai, China). Ammonia solution (NH_3_·H_2_O, 25%–28%) and sodium hydroxide (NaOH) were purchased from Sinopharm Chemical Reagent Co., Ltd. (Beijing, China). All the relative chemicals were directly used without further purification. Deionized (DI) water was used throughout the experiments.

### 2.2. Preparation of W-CDs

To prepare white emissive W-CDs, AA (1.0 g) was put into 20 mL DI water and 1.0 mL of ammonia solution added. The mixture was treated ultrasonically until a clear solution was obtained. Then the solution was transferred into a glass beaker, put into a domestic microwave oven and heated under 750 W irradiation. The microwave reaction continued until the clear, colorless solution completely turned into yellow-brownish, foam-like solid bulks. An appropriate amount of water was poured into the beaker to dissolve the obtained solids. The product solution was found to be weakly acidic due to consumption of ammonia during the reaction. A small quantity of NaOH solution was added to neutralize the crude product solution which was then subjected to dialysis (MWCO = 1000) against DI water for 2–3 days for the purpose of purification. The solid-state W-CDs can be obtained through freeze-drying. In the control experiments, glutamic acid (replacing AA) with the addition of ammonia and AA with the addition of NaOH (replacing ammonia) were performed under the same conditions for the preparation of W-CDs.

### 2.3. Procedures for Preparation of Emitting Films and W-LEDs

To prepare emitting films, the W-CDs were thoroughly mixed with epoxy resin at mass ratios of 1:250, 1:100, 1:50, 1:25, and 1:10, respectively, placed into a mold and then dried under ambient conditions. The emitting films could be obtained after being carefully peeled off from the mold.

Fabrication of W-LEDs: LED chips with UV emission centered at 365 nm were purchased from Shenzhen Looking Long Technology Co., Ltd (Shenzhen, China). The W-CDs and epoxy resin mixture were prepared with different mass ratios and utilized as light converting layers of a UV LED chip. After the addition of curing agent and a certain time of stirring and standing, the mixture was tipped over chips keeping a fixed volume. Then the chips were placed under ambient conditions until the curing process was complete. A working voltage of 3.0 V was applied to drive these LEDs.

### 2.4. Equipment and Characterization

The W-CDs were prepared using a domestic microwave oven (Media). Transmission electron microscopy (TEM) images were acquired on a Tecnai F20 electron microscope (FEI Company, Hillsborough, OR, USA). XPS data were produced with an ESCALAB 250Xi (Thermo Scientific, Waltham, MA, USA). Fourier transform infrared (FT-IR) spectra were measured on a Nicolet 6700 FT-IR spectrometer (Thermo Nicolet Corp., Madison, WI, USA) with the KBr pellet technique. Fluorescence emission and excitation spectra were obtained on a Hitachi F-4600 spectrophotometer equipped with a Xe lamp (150 W) under ambient conditions. UV-Visible absorption spectra were obtained on a PERSEE T10CS UV-Vis spectrophotometer (Persee, Beijing, China). PL photographs were taken using a Canon camera (EOS 550) under excitation by a hand-held UV lamp (365 nm). PL lifetime was measured on a HORIBA FL3-111 fluorescence spectrometer (HORIBA Jobin Yvon, Kyoto, Japan). PL QY measurements were carried out with a QE-2100 quantum efficiency measurement system (Otsuka Electronics, Tokyo, Japan). The EL spectra and device parameters of W-LEDs were collected from a PMA-12 spectrometer (Hamamatsu Photonics, Hamamatsu, Japan) equipped with a fiber integration sphere.

## 3. Results and Discussion

### 3.1. Preparation and Characterizations of W-CDs

The W-CDs were prepared by a facile and quick microwave-assisted method by heating an aqueous solution of AA in the presence of ammonia. After dialysis purification and freeze-drying, a yellow-brownish and fluffy powder could be obtained as the final product (please refer to the Experimental section for details). To demonstrate the critical role of AA and ammonia, two control experiments were performed, i.e., glutamic acid (having a similar molecular structure to AA) and a strong base (NaOH) being used to replace AA and ammonia, respectively. The corresponding results showed that neither of the above cases can produce solid-state white emissive CDs, indicating that both AA and ammonia are necessary for the successful preparation of the self-quenching- resistant W-CDs.

Transmission electron microscopy (TEM) measurements were first carried out to confirm microscopic morphology of the W-CDs. As shown in Figure 1a, the W-CDs are observed to be quasi-spherical shapes with an average diameter of 3.45 nm. The high resolution TEM (HR-TEM) image further confirms their amorphous feature (no lattice fringes can be recognized) (Figure 1b), implying a polymer-like structure nature. To demonstrate the possible chemical composition and structure of the W-CDs, X-ray photoelectron spectroscopy (XPS) and Fourier transform infrared (FT-IR) characterizations were performed. As depicted in Figure 1c, the full-scan XPS spectrum indicated the W-CDs mainly contained C (284.8 eV), N (399.7 eV), and O (530.8 eV) elements with relative molar contents of 55.77%, 14.06%, and 30.17%, respectively from which one can easily identify that the ratio between C and O in the W-CDs (1:0.54) is much higher than that of the starting material (i.e., AA, with C:O molar ratio to be 1:1). This indicates that a dehydration reaction must have occurred during the formation process of the W-CDs. Moreover, high resolution XPS C 1s, N 1s and O 1s spectra and their deconvoluted fitting results are shown in Figure 1d and Appendix A (Appendix A). Appendix A summarizes the relative contents of the various chemical bonds. From these results, the W-CDs could be determined to be mainly composed of amide bonds (O=C–N, 287.9 eV in C1s, 399.7 eV in N1s and 531.4 eV in O1s), amino groups (C–N–H, 286.0 eV in C1s and 401.5 eV in N1s), C–C/C=C bonds (285.3 eV in C1s), and hydroxyl groups (C–O–H, 284.5 eV in C1s and 533.1 eV in O1s) [50,51,52]. Thus, the occurrence of the condensation reaction could be confirmed by the formation of amide bonds in the W-CDs.

The FT-IR spectrum in Figure 1e displays strong and broad stretching vibrations at about 3200–3500 cm^−1^, intense and medium absorptions at 1712 and 1650 cm^−1^, arising from –OH/–NH_2_, carboxyl and amide carbonyl groups, respectively. The absorption peaks at 1352 and 1020–1250 cm^−1^ can be attributed to C–N and C–O stretching vibrations [53,54]. In short, these TEM, XPS, and FT-IR results demonstrated that the W-CDs could be polymer-like nanoparticles containing abundant hydroxyl, amino, carboxyl, and amide functional groups on their surface.

### 3.2. Optical Properties of the W-CDs

Similar to many other reported CDs, dilute dispersion of the W-CDs (in water) emits intense blue PL under UV excitation and exhibits excitation wavelength dependence, that is, the emission peaks red-shift with increasing excitation wavelength from 300 to 460 nm (Appendix A), and such a phenomenon has frequently been ascribed to the presence of multiple emissive centers/states in the CDs [4,5,6,55,56]. The UV-Vis absorption spectrum of W-CDs aqueous dispersion shows a major peak and a relatively weak shoulder peak in the UV region, and their peak positions can be readily determined to be 298 nm 352 nm by the fitting curves (Figure 2a). By reference to the literature, the two absorptions could be attributed to π→π* and n→π* transitions of the W-CDs, respectively [25,57]. In addition, the W-CDs were observed to exhibit two excitation peaks at about 295 and 350 nm based on their PL excitation spectrum (Figure 2a), which are in good accordance with their absorption maxima, indicating that the PL emission could have arisen from the two absorption relevant moieties contained in the W-CDs.

Interestingly, unlike many other reported CDs, which usually suffer quenching in the solid state, the W-CDs powder displays unexpected white PL emission under 365 nm UV-lamp irradiation (Figure 2b). To further demonstrate the PL properties, emission spectra of the W-CDs powder were measured. As shown in Figure 2c, the W-CDs powder exhibits a feature of remarkable excitation-wavelength dependence with broad PL spectra under excitation from 300 to 400 nm and sing-peak emissions from 420 to 480 nm. Specifically, the PL spectrum is found to cover nearly all the visible region from 400 to 700 nm by 365 nm excitation (Figure 2c), being responsible for their white light emittance. Note that the white light emission of W-CDs powder under 365 nm irradiation presents a Commission Internationale de l’Eclairage (CIE) 1931 coordinate of (0.31, 0.35), which is very close to the coordinate (0.33, 0.33) of pure white light. According to the 2-dimensional (2D) excitation-emission matrix of W-CDs, the PL spectrum was found to mainly be comprised of two emissive components, one locating at the blue-region and the other at the green to orange regions (Figure 2d). Such a unique dual-emission PL characteristic makes the W-CDs powder an ideal candidate to fabricate single-component-based W-LEDs.

To obtain a more in-depth understanding of why the W-CDs exhibit blue PL emission in dilute dispersion but emit white light in the solid state, their concentration-dependent PL property in aqueous dispersion was examined. With concentrations of the W-CDs increasing from 0.5 to 32 mg mL^−1^, their PL spectra were observed to slightly red-shift but remarkably broaden with a new emission emerging at longer-wavelength region (Figure 3a), accompanied by emission colors changing from blue to bluish green and to bluish white (Figure 3b). Consequently, the newly emerged longer-wavelength emission at the yellow to orange regions of the W-CDs in the solid state could be ascribed to their aggregation, and such a property has been identified as one of the characteristics of solid state emissive CDs due to re-absorption and RET effects [25,34]. In addition, it is worth noting that the PL QYs of W-CDs aqueous dispersion were found to decrease with increasing concentration (Figure 3b) and finally reach the lowest in the solid form. Although a relatively large reduction of PL QYs from dilute dispersion (23.4%) to solid (6.7%), the W-CDs powder still represents the highest QY value among the reported single-component solid white emissive CDs.

### 3.3. Possible Formation Process and PL Mechanism of the W-CDs

Based on the above discussion, the possible formation process and PL mechanism of the W-CDs are proposed as follows: first of all, dehydration and condensation reactions between amino and carboxyl groups on AA occur and result in cross-linked polymers under the microwave-irradiation heating condition, while the presence of ammonia is thought to facilitate these reactions; then, part of the polymers undergoes carbonization to form more dense carbon cores with the rest wrapped on their surface (Scheme 1). In addition, according to the chemical composition and structure analyses, the W-CDs are confirmed to contain abundant sub-fluorophores, such as –NH_2_ and C=O, and their PL emission is thus considered to have resulted from the CEE of these sub-fluorophores due to the cross-linked polymer structure nature of the material. In fact, such a PL emission mechanism has been widely reported and accepted for these kinds of structured CDs [31]. As to the self-quenching-resistant PL feature of the W-CDs, it is tentatively attributed to their “core-shell” structure. The polymer-like shell layer could separate cores at a certain distance, and this reduced, at least partially, the energy transfer process among the W-CDs that being regarded as a major reason for self-quenching. Nevertheless, the QYs of the W-CDs are observed to be lower in the solid state (6.7%) than in the dilute dispersion state (23.4%), indicating part of the self-quenching still exists in their solid form. Based on these discussions, effectively white light emission from the W-CDs powder could be attributed to their blue emission in the dispersion state, the self-quenching-resistant feature, and the re-absorption-/RET-induced long-wavelength yellow to orange emissions in the solid state. Almost all the other reported CDs, however, do not possess such properties and therefore solid white light emissive CDs had actually been achieved very successfully [58,59].

To further demonstrate the PL mechanism of the W-CDs, their PL decay curves in the solid and dispersion states were measured. As shown in Figure 4 and Appendix A, both of the two cases exhibited bi-exponential decay PL behaviors, implying two-component-based emission processes. In the dispersion state, the two emissions may have arisen as they contained different sub-fluorophores in the W-CDs (e.g., NH_2_ and C=O). In the solid state, however, the two emissions could be tentatively attributed to their isolated and aggregated forms, respectively. Moreover, the average lifetime in the dispersion state (5.92 ns) was determined to be obviously larger than that in the solid state (2.36 ns) (Appendix A), further supporting the occurrence of RET among W-CDs because of its competition with radiative transition and consequently shortening of the lifetime [25]. Besides, the excitation dependence of W-CDs is possibly another key factor for the appearance of longer wavelength emissions (i.e., yellow to orange regions) in high concentrations of dispersion and solid state due to the re-absorption process [60].

### 3.4. Applications of the W-CDs in Light-Emitting Films and W-LEDs

The unique concentration-dependent PL properties in the dispersion state and the white light emission in the solid state make the W-CDs applicable for various solid-state lighting fields, such as light-emitting thin films and white lighting [61]. First of all, considering the concentration-dependent PL feature, various mass ratios of the W-CDs and epoxy resin were mixed to prepare solid-state light emitting thin films, in which the W-CDs act as color converting dopants and epoxy resin as the host matrices, respectively. As shown in Appendix A, the obtained films emit blue, bluish white, and white PL under UV-lamp (365 nm) irradiation with the concentrations of W-CDs increasing from 0.4% to 10% (mass ratio), attributed to the gradual appearance and enhancement of the new long-wavelength emission at yellow to orange regions (Appendix A). These results are also reflected in the shifts of CIE 1931 chromaticity coordinates along the direction from blue to yellow (Appendix A). To evaluate the stability of the W-CDs, PL spectra of a light emitting thin film (the 10% mass ratio material contained one being taken as an example) were measured under various periods of UV irradiation (365 nm, 150 W Xe lamp). As shown in Appendix A, only a minor decline of the PL emission of the thin film was observed, i.e., more than 96% emission intensity was still retained after 3 h of continuous irradiation, indicating very good stability of the W-CDs.

Next, W-LEDs were attempted to be fabricated by packaging various mass ratios of the W-CDs and epoxy resin (one of the frequently used packaging materials in fabricating LEDs) on commercially available UV chips (365 nm) and the corresponding light-emitting performance is shown in Figure 5a and Table 1. Similar to the light-emitting thin films, the electroluminescence (EL) spectra of these LEDs exhibit more intense emission at long-wavelength regions with the concentrations of W-CDs increasing (Appendix A). Based on the EL spectra, the CIE 1931 chromaticity coordinates of these LEDs can be determined and were found to shift from near white region to orange-yellow regions (Figure 5b). It is worth noting herein that a much lower ratio of the W-CDs was used to obtain near white light emission in LEDs than that in thin films. This is reasonably ascribed to the fact that the W-CDs and epoxy resin composite on a UV chip could induce more intense long-wavelength emission due to a more effective re-absorption effect (Figure 5c). As a specific example, with only 1% mass ratio of the W-CDs being used, the as-fabricated LED exhibits superior white lighting performances (Figure 5d and Table 1), including luminance of 6002 cd·m^−2^, luminous efficiency of 1.281 lm W^−1^, CRI of 83, CCT of 6987 K, and chromaticity coordinates of (0.30, 0.35), which is very close to the pure white light coordinates of (0.33, 0.33). These white lighting parameters are found to be obviously enhanced in comparison to the reported single-component CDs based W-LEDs (Appendix A) [29,53]. To make a direct evaluation, the as-fabricated warm white W-LED (with 10% mass ratio of the W-CDs) displays satisfactory performance in a practical lighting application (Figure 5e). Overall, the above discussion demonstrates that the W-CDs could be employed in fabricating single-component-based W-LEDs with advantages of UV-light-excitable and tunable CCT features (from cool to warm white light) by simply increasing their concentrations relative to the packaging materials (e.g., epoxy resin). Note that, in comparison with the traditional rare-earth phosphors for fabricating LEDs, the W-CDs exhibit merits of lower cost and energy consumption in preparation, less deleterious to the human and the eco-system, and a more facile process in fabrication, but they currently cannot compete with the rare-earth based phosphors in stability and PL QYs in the solid state.

## 4. Conclusions

In summary, a facile, rapid, and scalable method for the synthesis of solid-state white emissive CDs is reported in this study. The as-prepared W-CDs exhibit blue PL in dilute aqueous dispersion but unique white light emission in the solid state under UV excitation. Furthermore, the possible chemical structure, the formation process, and the PL mechanism of the W-CDs in dispersion and solid state were discussed. Thanks to the unique optical properties, W-CDs were employed to fabricate single-component-based W-LEDs by using commercially available UV chips which exhibited satisfactory performances, including high white light-emitting purity, high CRI, and tunable CCT, rendering great promise for the W-CDs in the field of white lighting. Finally, it is necessary to point out that the PL QY of the W-CDs powder is still relatively low in comparison to the traditional rare-earth phosphors. The following work should be focused on improving PL QYs of white solid-state emissive CDs and such work is now in progress in our laboratory.

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
