# Peer review of "A Facile Approach to Solid-State White Emissive Carbon Dots and Their Application in UV-Excitable and Single-Component-Based White LEDs"

_nanomaterials, 2019, doi:10.3390/nano9050725_

Round 1

Reviewer 1 Report

This paper describes the preparation of solid-state white emitting carbon dots (CD). Although this subject is not very innovative, the paper is well structured, the material well characterized and the results were properly discussed. Some further discussion, in comparison with others, should be done on why this sample of CD emits white light in the solid phase while other don’t. Also, another point should be further discussed, which is related with the comparison of the proposed CD with the current used RE phosphors in LEDs production. Indeed, this is an important information for the reader to understand the real interest of these new solution in comparison with the currently used. 

Author Response

Responses to the reviewer 1

This paper describes the preparation of solid-state white emitting carbon dots (CD). Although this subject is not very innovative, the paper is well structured, the material well characterized and the results were properly discussed. Some further discussion, in comparison with others, should be done on why this sample of CD emits white light in the solid phase while other don’t. Also, another point should be further discussed, which is related with the comparison of the proposed CD with the current used RE phosphors in LEDs production. Indeed, this is an important information for the reader to understand the real interest of these new solution in comparison with the currently used.

Response: Thanks the reviewer’s suggestion. Some further discussion about “why this sample of CD emits white light in the solid phase while other don’t” had been added in the revised manuscript (page 6, line 236-240), which can also be seen as follows: “Based on these discussion, effectively white light emission from the W-CDs powder could be attributed to their blue emission in dispersion state, self-quenching-resistant feature and re-absorption-/RET-induced long-wavelength yellow to orange emissions in the solid state. Almost all the other reported CDs, however, do not possess such properties and therefore the solid white light emissive CDs had actually very severely been achieved [57,58].”

In addition, “the comparison of the proposed CD with the current used RE phosphors in LEDs” had also been added in the revised manuscript (page 8, line 295-299), which can be seen as follows: “Note that, in comparison with the traditional rare-earth phosphors for fabricating LEDs, the W-CDs exhibit merits of lower cost and energy consumption in preparation, less deleterious to human and eco-system, and more facile process in fabrication, but they currently cannot compete with the rare-earth based phosphors in stability and PL QYs in solid state.”

Reviewer 2 Report

The authors report the preparation of solid-state white emissive CQDs using L-aspartic acid and ammonia as precursors. CQDs were demonstrated to emit blue light when dispersed in aqueous solution and white light in the solid state . The CQDs were successfully used for the preparation of white LEDs. The results are of interest and clearly presented. The following comments should be considered by the authors :

- some recent papers related to the use of CQDs for LEDs applications should be added  (Nanoscale 2016, 8, 8618-8632; Nature Communications 2018, 9, 2249).

- line 68 : All SQDs using for LEDs don't contain toxic heavy metals as stated by the authors (see for example reference 9). This should be corrected in the text.

- did the authors try to prepare CQDs from glutamic acid and ammonia ?

- XPS : it would be of interest for the readers to provide the high resolution spectra of N 1s and O 1s and to discuss the results.

- the excitation wavelength dependent PL emission of CQDs is extensively documented in the literature. A reference should be added and the results discussed.

- from my opinion, the PL lifetime paragraph should be improved.

- it would be of interest to evaluate both the thermal and the long-term stability of W-LEDs prepared with CQDs to demonstrate the potential of CQDs in LEDs applications. The performances of the LEDs should also be compared to those described in the literature.

Author Response

Responses to the Reviewer 2

The authors report the preparation of solid-state white emissive CQDs using L-aspartic acid and ammonia as precursors. CQDs were demonstrated to emit blue light when dispersed in aqueous solution and white light in the solid state. The CQDs were successfully used for the preparation of white LEDs. The results are of interest and clearly presented. The following comments should be considered by the authors:

- some recent papers related to the use of CQDs for LEDs applications should be added  (Nanoscale 2016, 8, 8618-8632; Nature Communications 2018, 9, 2249).
   Response: The two papers that the reviewer mentioned had been added and cited in the revised manuscript (i.e., Ref. 3 and 27).

- line 68 : All SQDs using for LEDs don't contain toxic heavy metals as stated by the authors (see for example reference 9). This should be corrected in the text.
   Response: Thanks the reviewer pointed out this mistake for us and relevant references had been corrected and updated. In details, Refs. 9 and 10 were cited to show the applications of rare-earth phosphors, which generally don’t contain toxic heavy metals. In addition, several SQDs based phosphors that applied for LEDs were cited (Refs. 11-13, 39 and 40) in the revised manuscript and all of them contain toxic heavy metals (e.g., Pb or Cd).

- did the authors try to prepare CQDs from glutamic acid and ammonia?
   Response: Yes, we did try to prepare CDs using glutamic acid and ammonia as starting materials and the relevant discussion was actually had been provided at the end of the first paragraph in Section 3.1 (page 3, line 135-140). Such discussion can be seen as follows as well: “To demonstrate the critical role of AA and ammonia, two control experiments had been performed, i.e., glutamic acid (having a similar molecular structure to AA) and a strong base (NaOH) being used to replace AA and ammonia, respectively. The corresponding results exhibited that neither of above cases can produce solid-state white emissive CDs, indicating that both of AA and ammonia are necessary in successful preparation of the self-quenching- resistant W-CDs.”

- XPS : it would be of interest for the readers to provide the high resolution spectra of N 1s and O 1s and to discuss the results.
   Response: As the reviewer’s suggestion, high resolution XPS spectra of N1s and O1s were provided (Figure S1 in SM) and the results were discussed in the revised manuscript (page 4, line 152-159). The relevant discussion can also be seen as follows: “Moreover, high resolution XPS C1s, N1s and O1s spectra and their deconvoluted fitting results were shown in Figure 1d and Figure S1 (Supplementary Materials, SM). Table S1 in SM summarized the relative contents of various chemical bonds. From these results, the W-CDs could be determined mainly composing of amide bonds (O=C-N, 287.9 eV in C1s, 399.7 eV in N1s and 531.4 eV in O1s), amino groups (C-N-H, 286.0 eV in C1s and 401.5 eV in N1s), C-C/C=C bonds (285.3 eV in C1s), and hydroxyl groups (C-O-H, 284.5 eV in C1s and 533.1 eV in O1s) [51-53]. Thus, the occurrence of condensation reaction could be confirmed by the formation of amide bonds in the W-CDs.”

- the excitation wavelength dependent PL emission of CQDs is extensively documented in the literature. A reference should be added and the results discussed.
   Response: References about the excitation wavelength dependent PL emission of CDs had been added in the revised manuscript (i.e., Refs. 4-6, 55 and 56), and a brief discussion was provided as well (page 4, line 172-175). The relevant discussion can also be seen as follows: “…the emission peaks red-shift with excitation wavelength increasing from 300 to 460 nm (Figure S2, SM), and such a phenomenon has frequently been ascribed to the presence of multiple emissive centers/states in the CDs [4-6,55,56].”

- from my opinion, the PL lifetime paragraph should be improved.
   Response: According to the reviewer’s suggestion, the PL lifetime paragraph had been appropriately improved in the revised manuscript (page 7, line 244-248), which can also be seen as follows: “As shown in Figure 4 and Table S2 (SM), both of the two cases exhibited bi-exponential decay PL behaviors, implying two-component-based emission processes. In the dispersion state, the two emissions may be arisen from containing different sub-fluorophores in the W-CDs (e.g., NH2 and C=O). In the solid state, however, the two emissions could be tentatively attributed to their isolated and aggregated forms, respectively.”

- it would be of interest to evaluate both the thermal and the long-term stability of W-LEDs prepared with CQDs to demonstrate the potential of CQDs in LEDs applications. The performances of the LEDs should also be compared to those described in the literature.
   Response: We agreed with the reviewer that thermal and long-term stability are very important for the potential applications of LEDs. To evaluate stability of the W-CDs, PL spectra of a light emitting thin film (with 10% mass ratio of the W-CDs being taken as an example) were measured under various periods of UV irradiation (365 nm, 150 W Xe lamp). As shown in Figure S4 (SM), only minor declines of the PL emission of the thin film were observed, i.e., more than 96% emission intensity still retained after 3 hours of continuous irradiation, indicating good photo-stability of the W-CDs and the emitting thin film. Unfortunately, the EL emission of the W-LED (containing 1% mass ratio of the W-CDs being taken as an example) was observed to be gradually decreased with prolonging time of lighting (Figure R1). This observation might be attributed to the method that we fabricated the W-LEDs, which induced rapid temperature enhancement of the UV chip due to poor heat dissipation of the device. Since the W-CDs exhibited very good photostability, we believe that the thermal and long-term stability of corresponding LEDs should not be a problem if the heat dissipation is solved by appropriately fabricated LEDs. Of course, such tests have to be performed and we are trying on this issue in our lab.
In addition, the performances of the as-fabricated LEDs and those described in the literature were compared and relevant results were summarized in Figure S3 (SM).

Figure R1. EL spectra of the W-LED with 1% mass ratio of the W-CDs after various periods of lighting.

Round 2

Reviewer 2 Report

All corrections were done by the authors. I recommend this manuscript for publication in Nanomaterials.